# Neuropsychiatric Disorders in Farmers Associated with Organophosphorus Pesticide Exposure in a Rural Village of Northwest México

**DOI:** 10.3390/ijerph16050689

**Published:** 2019-02-26

**Authors:** Aracely Serrano-Medina, Angel Ugalde-Lizárraga, Michelle Stephanie Bojorquez-Cuevas, Jatniel Garnica-Ruiz, Martín Alexis González-Corral, Arnold García-Ledezma, Gisela Pineda-García, José Manuel Cornejo-Bravo

**Affiliations:** 1Facultad de Medicina y Psicología, Universidad Autónoma de Baja California, Calzada Universidad 14418, Parque Industrial Internacional, Tijuana 22300, Mexico; serrano.aracely@uabc.edu.mx (A.S.-M.); ugalde.angel@uabc.edu.mx (A.U.-L.); jgarnica@uabc.edu.mx (J.G.-R.); alexis.gonzalez@uabc.edu.mx (M.A.G.-C.); arnold.garcia@uabc.edu.mx (A.G.-L.); gispineda@uabc.edu.mx (G.P.-G.); 2Facultad de Ciencias Químicas e Ingeniería, Universidad Autónoma de Baja California, Calzada Universidad 14418, Parque Industrial Internacional, Tijuana 22300, Mexico; michelle.bojorquez@uabc.edu.mx

**Keywords:** acetylcholinesterase, mental disorders, neurotoxicity, organophosphates, pesticides

## Abstract

This study aims to determine the degree of acetylcholinesterase inhibition and neurological symptoms for each of the psychiatric disorders diagnosed in the farm workers of a rural population in the state of Baja California, Mexico. We conducted a cross-sectional study on 140 agricultural workers (exposed participants). The study was run using the Mini International Neuropsychiatric Interview Diagnostic Test (MINI), a pre-established questionnaire to diagnose the mental state of each agricultural worker. Analysis of enzymatic activity was carried out using the modified Ellman method. The results showed that, among agricultural workers with slightly inhibited enzymatic activity, 25% met the criteria for the diagnosis of major depression with suicidal attitudes, 23.9% with inhibited enzymatic activity showed generalized anxiety, 23.5% showed combined depression–anxiety, and 22% met the criteria for major depression and no psychiatric diagnosis disorder. These results suggest the need for the development of effective public-health strategies to inform farm workers about integrated pesticide management in order to prevent serious health complications.

## 1. Introduction

Exposure to organophosphates (OPs), such as nerve agents and pesticides, results in an irreversible inhibition of acetylcholinesterase (AChE) in neuromuscular junctions, peripheral and central neuronal synapses, and in circulating red blood cells [1]. Inhibition of AChE, which catalyzes the hydrolysis of the acetylcholine (ACh) neurotransmitter, results in increased levels of synaptic ACh and subsequent overstimulation of nicotinic and muscarinic ACh receptors. Cholinergic toxicity is clinically manifested by decreased heart and respiration rates, increased secretions, muscle tremors, paralysis, and death [2,3]. Moreover, AChE is the primary target of organophosphorus and carbamate pesticides, the most commonly used classes of insecticides worldwide [4]. Apart from their lethal acute toxicity at high doses, the extensive use and availability of OPs in agricultural and domestic applications raise questions about the safety of long-term exposure at the currently approved levels [5,6,7]. Furthermore, there is research that correlates the environment of rural populations with the inhibition of acetylcholinesterase enzymatic activity, which supports the idea that exposure to OPs may cause depression disorders with a tendency toward suicide [8]. There are several studies on pesticide exposure as a possible risk for stimulating the onset of diseases such as Parkinson’s disease, Alzheimer’s disease, attention-deficit hyperactivity disorders, anxiety disorders, and depression in underdeveloped countries, especially in rural populations [9,10,11]. As for suicide, studies show higher rates in areas with intensive use of pesticides compared to areas with less use of pesticides. Occupation in agriculture has also shown significant association with an increased risk of suicide compared with other occupational groups. Nevertheless, scientific evidence is still limited and inconclusive in some populations [12].

The state of Baja California, Mexico, constitutes a pole of attraction for population from the rest of the country, partly due to its economic dynamics and geographical location close to the United States market [13]. Agricultural activities are intensive in the Mexicali and San Quintin Valleys, with high market value [14,15]. Intensive agricultural activities require the use of pesticides that are, undeniably, highly toxic, absorbable, and persistent, representing a health risk to farm workers. The use of OP pesticides in the northwest of the country favors products that contain insecticides, such as malathion, dicofol, methyl parathion, chlorpyrifos, phosmet, azinphos-methyl, and methamidophos [13,16].

Exposure to pesticides is not limited to workers who are directly exposed, but also includes nearby workers, their families, and residents near the cultivation areas. Therefore, farm workers are often doubly exposed, both environmentally and occupationally, since it is common in this sector to live very close to cultivation areas. For this reason, exposure becomes chronic. This study aims to determine the degree of acetylcholinesterase inhibition and neurological symptoms for each of the psychiatric disorders diagnosed in the farm workers of a rural population in the state of Baja California, Mexico, with the purpose of establishing the relationship between pesticide-caused neurotoxicity and psychiatric disorders. It is necessary to investigate all these aspects in order to carry out programs that can improve the health of farm workers, and also to develop a monitoring system for psychiatric disorders, which must be available when farmers have neurological symptoms from pesticide exposure.

In this study, we use the modified Ellman test by Worek et al. (1999), which provides a simple method for the sensitive and accurate determination of AChE activity in whole blood in the presence of organophosphates and oximes, with reasonable reproducibility (coefficient of variation below 3%) [17].

## 2. Materials and Methods

### 2.1. Study Design and Participant Selection

A community-based cross-sectional study was designed across three time periods—May 2017, October 2017, and May 2018—in the San Quintin Valley, Baja California, in Northwest Mexico. A total of 140 participants living in the district of Venustiano Carranza were selected. The participants are dedicated to agricultural work involving pesticide application, vegetable collection in areas previously sprayed with pesticides, and packing of vegetables. The selected participants were vegetable collectors over 18 years old, of both sexes, residing in the geographical area of sampling for at least 6 months. For the control population, 100 nonexposed participants from an urban-zone population were selected, for convenience, in the same periods as the exposed participants. Individuals under 18 years of age and with less than six months of residence in the study area were excluded.

### 2.2. Ethical Statement

This study was conducted according to the Declaration of Helsinki, and the protocol was approved by the Ethics Committee of the Faculty of Medicine and Psychology of the Autonomous University of Baja California (FMP-PI-2017-005). Participants received explanations about the study and its purpose in Spanish. Informed consent was obtained from research participants before the start of each interview. We prepared written and verbal informed consent, which included the purpose of the investigation, the expected duration of the interview, and a declaration that the participants would be at no risk because of the study, could withdraw from the interview at any time, and would receive no payment for their participation. We read this statement to each study participant during the process of requesting their permission to participate in the study. The signed consent forms were stored by the main author of the study.

### 2.3. Data Collection

A trip was made to the Venustiano Carranza district of San Quintin Baja California, Mexico, to collect data and blood samples. The Mini International Neuropsychiatric Interview Diagnostic Test (MINI) was conducted with the farm workers exposed to the pesticides to assess their mental-health status. For nonexposed participants, the MINI was applied under the same conditions. A total of three medical students, previously trained on the importance of information confidentiality, were the data collectors and responsible for administering the MINI. A psychologist assessed the mental-health status based on the test results. The MINI is a short diagnostic interview that can easily be incorporated into routine clinical interviews; it is a detection instrument and diagnostic orientation in the discipline of psychiatry and psychology. It is a feasible, acceptable, and reliable test–retest [18,19]. The MINI is based on the Diagnostic and Statistical Manual of Mental Disorders (DSM-IV) published by the American Psychiatric Association [20]. We collected information on the sociodemographic status, economics, health, and symptomatology of those who had experienced exposure to pesticides. It was not possible to collect data on the use of pesticides on farmland, since farm workers do not know what type of mixture they use to manage pests in agricultural fields. However, the data regarding pesticide use were taken from the Ministry of Agriculture, Livestock, Rural Development, Fisheries and Food Supply’s (SAGARPA, by its initials in Spanish) Agricultural Technical Agenda [13,16].

### 2.4. Sample Preparation

Before the study, a laboratory was set up in the town’s medical clinic to obtain and perform the enzymatic tests. All exposed participants had previously given their informed consent and their mental health had been evaluated before they proceeded to give a blood sample by venous puncture. Two biochemists measured AChE activity using the colorimetric modified Ellman assay (Worek method) [17,21]. The vacutainer system was used to take 3 mL of venous blood in tubes with K_2_EDTA anticoagulant. Anticoagulated blood was separated into plasma and erythrocytes by centrifugation for 10 min at 3000 rpm. Red blood cell samples were washed 3 times in cold isotonic saline (0.9%, *v*/*w*) and then hemolyzed by centrifuging for 10 min at 14,000 rpm. All collected samples were immediately processed to determine acetylcholinesterase activity according to the Worek method [17]. The only difference in this study compared with that referencing the Worek method was the immediate analysis of fresh blood extracted from the participants; thus, we did not require or use any process for storing the samples.

### 2.5. Determination of AChE Activity

Blood dilution (1:100, prepared with Triton X-100 (Sigma-Aldrich, Toluca, Estado de Mexico, Mexico), 1 mL of phosphate buffer (0.1 M, pH 7.4), and 0.05 mL of DTNB (5,5’-dithiobis 2-nitrobenzoic acid, 10 mM) were added together. The mixture was incubated at 37 °C for 10 min, and then 0.025 mL of acetylthiocholine iodide (28.3 mM) was added. The absorbance was monitored at 436 nm for 3 min using a UV/Vis spectrophotometer (DU-50 Beckman Coulter, Carlsbad, CA, USA). Hemoglobin concentration was determined from the hemolyzed erythrocytes using a hematology analyzer (Advia60, Bayer, Leverkusen, Germany). The amount of hydrolyzed substrate was correlated with hemoglobin concentration levels to calculate the AChE specific activity, and was expressed as mU/μLHb. The reproducibility of enzymatic analyses was assessed by triplicate analysis of blood samples. In each case, the variation coefficient was 8% or less. To determine the activity of acetylcholinesterase (mE/min), the linearity of the Worek method test was taken [17]. Linear regression analysis resulted in a high correlation (*r*^2^ = 0.9975, *y* = 52.3536*x* − 1.5994). We were able to determine the percentage of inhibition by using the same method. 

### 2.6. Data Analysis

SPSS software package 21.0 (IBM, Chicago, IL, USA) was used for all statistical evaluations. Descriptive statistics included mean, median, standard deviations, and range values for continuous data; for categorical data, they included percentage and frequency. Comparisons between groups were analyzed using the Student’s *t*-test, Chi-square test, and ANOVA, depending upon the distribution of variables. Correlation analysis was performed with the Spearman correlation test. Binary logistic regression analysis was undertaken for the evaluation of different variables on suicide risk. Statistical significance was accepted as *p* < 0.05.

## 3. Results

The sociodemographic characteristics of the exposed participants included an average age of 32.4 years, with 62.9% being women. The nonexposed participants’ average age was 23.4 years, with 60.0% being women. The distribution of age, gender, education, marital status, and unhealthy habits for the study population is summarized in Table 1. Significant differences in sociodemographic variables were observed between the exposed and nonexposed groups, except in the sex and alcohol consumption of the participants. Among the exposed participants, 55.7% had an elementary-school education, and 27.1% had a middle-school education, while the highest percentage in the nonexposed participants had high-school-level education (97%). Regarding the diagnoses of psychiatric disorders, Table 1 also shows the results of the MINI interviews with farm workers exposed to pesticides in terms of their mental-health status, as well as those of the nonexposed participants. Significant differences were observed in the diagnosis of psychiatric disorders between exposed and nonexposed participants (*p* < 0.001): 31.4% of the exposed participants had a diagnosis of major depression with suicidal risk; the diagnoses of major depression, generalized anxiety, and depression–anxiety were found to have equal percentages (14.3%), and no psychiatric disorders were detected in 25.7% of the exposed group.

Regarding the nonexposed group, no psychiatric disorders were detected in 64% of the participants, followed by generalized anxiety (18%), and 8% of the participants presented with major depression with suicidal risk.

The evaluated hematological parameters in the study population are shown in Table 2. The average hemoglobin contents of female participants in both the exposed and nonexposed groups were considered as normal for a Mexican population, according to Diaz et al. [22]. Men from among the exposed participants had lower hemoglobin contents than the nonexposed men (*p* < 0.001).

The mean (standard error) levels of total acetylcholinesterase activity for exposed and nonexposed participants are shown in Table 2; the mean value of the enzymatic activity of AChE for the exposed population was significantly lower (178.88 mE/min ± 37.48) than the enzymatic activity of the nonexposed group (227.95 ± 10.19), (*F* = 126.9, *p* < 0.001). Therefore, the percentage of activity for the exposed participants was low compared to that of the nonexposed participants. It was also observed that the lower the percentage of activity was, the greater the percentage of enzymatic inhibition. Regarding the specific activity of AChE, significant differences were observed in both genders, but especially in men because of the low hemoglobin content (*F* = 33.2, *p* < 0.001). Farm workers exposed to pesticides showed lower specific activity or lower percentage of activity compared to nonexposed participants (*F* = 126.9, *p* < 0.001). The average value of the specific activity of AChE in the exposed group was 417.34 mU/μLHb—a notably low value according to Worek et al. [20].

According to that reported by SAGARPA [13], the most commonly used pesticides in Baja California México are OPs (Table 3), including pesticides belonging to the IA (extremely hazardous) or IB (highly hazardous) groups of the World Health Organization classification [23].

Regarding the risk of toxicity and neurological symptoms of the exposed participants (Table 4), none of the farm workers knew which pesticide was used, but they did know that they used more than one pesticide at a time, or a pesticide mix; moreover, more than 95% of farm workers reported not using any protective equipment while working in the field. Study participants worked 51.94 hours per week on the farmland (30 to 90 h per week) on average, and six days per week; farm workers had also been exposed to the pesticides they used for an average of 2.96 years (between 0.5 and 20 years).

Eighty percent of the participants in the exposed group had at least one neurological symptom, and many of the farmer workers reported having symptoms during the workday (Table 4), the most common of which included back pain (64.3%), numbness (51.4%), dizziness (51.4%), shoulder pain (48.6%), abdominal discomfort (40.7%), dyspnea (31.4%), nocturia (34.3%), and insomnia (25.7%). It could also be observed that the workers who were diagnosed with major depression with suicidal risk had a higher frequency of neurological symptoms when compared with workers who were not diagnosed with any psychiatric disorder. 

Student *t*-test (independent samples) factorial ANOVA shows results that were obtained for enzymatic activity, specific activity, percentage of activity, and the percentage of enzymatic inhibition (Table 5, Figure 1). It can be observed that exposed participants diagnosed with major depression with suicide risk had the highest percentage of inhibition (25.0%) in comparison with participants not exposed to pesticides (2.3%), so statistical analysis showed significant difference between the groups (*t* = 4.6, *p* < 0.001). The workers in the exposed population and those not exposed to pesticides, without psychiatric disorders, showed the lowest percentage of inhibition, and the groups demonstrated significant differences (*t* = 8.7, *p* < 0.001). The group of exposed workers diagnosed only with depression did not show significant differences in the enzymatic parameters compared with the nonexposed group (*t* = 2.0, *p* = 0.057). Nevertheless, the exposed participants showed the lowest enzymatic-activity percentage in both study groups.

In logistic regression models, the unadjusted-odds ratio for association between exposure to the pesticides and suicide risk was 5.32 (95% confidence interval, CI 2.37–11.93 and *p* < 0.001); a woman has 1.4 times the probability of a man to present with suicide risk (CI 0.738–2.83 and *p* = 0.283). The association between AChE activity and suicide risk was 0.490 (95% confidence interval, CI 0.993–0.999 and *p* = 0.006). A negative Spearman correlation was found between the frequency of suicide risk and AChE activity levels (*r* = −0.017, *p* = 0.915). The suicide risk of exposed workers was classified as high (*n* = 27, 63.36%), moderate (*n* = 9, 20.45%), or low (*n* = 8, 18.18%), and it was observed that AChE activity levels (416.11 ± 106.18 μU/μLHb) of those who had high suicide risk were lower than the levels of patients with low suicide risk (443.92 ± 84.01 μU/μLHb), but the difference was not statistically significant (*t*(33) = 0.677, *p* = 0.503).

## 4. Discussion

The results of the study provide some evidence of an association between chronic occupational exposure to OPs and neuropsychological effects such as depression, suicidal risk, and major anxiety. Pesticide exposure seems to affect not only the person spraying the pesticides, but also nonspraying workers as well [23]. When spraying insecticide in vegetable fields, farmers can be exposed to several kinds of insecticides, including OPs and carbamates [24]. These kinds of insecticides can be absorbed into the body via the gastrointestinal tract, by inhalation, and through the skin [25,26]. Overexposure to OPs causes phosphorylation of the serine in the active site of AChE, resulting in loss of enzymatic activity, followed by accumulation of the neurotransmitter acetylcholine (ACh) in the body, cholinergic overstimulation, and disorder of numerous bodily functions with adverse effects on health [4,10]. Worldwide, OPs are the most widely used class of pesticides [4,27]. Exposure to OPs may have chronic, long-term effects, and has been linked to delayed onset peripheral neuropathies primarily affecting the extremities, and causing neuropsychological and neurobehavioral changes [10,28].

In this study, there were differences between psychiatric symptoms and enzymatic activity, as well as differences in sociodemographic variables between exposed and nonexposed groups. It is important to consider that previous studies [29] have shown that economic, social, and demographic factors, as well as low levels of social support, also affect the symptoms of anxiety, depression, and suicidal ideation [30]. Nevertheless, it is increasingly suggested that exposure to OPs is a risk factor for depression and suicide. However, the epidemiological evidence is still very limited regarding the effects of chronic and low-dose exposure [8,12,31].

Magauzi et al. [32] evaluated the health effects of agrochemicals on agricultural workers in commercial farms in the Kwekwe District (Zimbabwe), as well as blood cholinesterase activity levels, which were 75% or less than normal levels. The median period of exposure to agrochemicals that they studied was three years. However, they did not precisely define the method they used to measure enzymatic activity, which must be very clearly established in a study of pesticide poisoning.

Cholinesterase activities, including butyrylcholinesterase (BChE) and AChE, serve as useful and sensitive biomarkers to monitor exposure to cholinesterase-inhibiting substances, such as chemical-warfare nerve agents and the agricultural use of pesticides [20,21,33,34].

Additionally, a key aspect is the suitable selection of biomarker type according to the occupational characteristics involved. The determination of AChE activity in red blood cells is a biomarker used to evaluate acute poisoning or chronic exposure to pesticides. BChE measurement is used only for acute poisoning diagnosis [35].

Regarding stability, native blood dilutions were stable for at least seven days (97.3%), slowly decreasing to 91.8% in 34 days, while the activity of BChE in plasma remained unchanged for 34 days [20]. Here, our methodology was modified due to loss of AChE activity of native blood dilutions after seven days of storage. We took into account this critical loss, so we decided not to store the samples at −20 °C. Instead, we processed the sample immediately after the venous puncture to the patient to instantaneously quantify the activity. The Worek method also makes reproducible measurements in the presence of a possible inhibitor, even with low-tech equipment, and can serve as a tool to detect whether people have been exposed to organophosphates, and to identify patients intoxicated with organophosphates.

The exposed and nonexposed groups present differences in confounders such as age, education and marital status (Table 1). These differences preclude to draw conclusions about differences between groups regarding the diagnosis of psychiatric disorders. However, there is a trend in the results: at least 31.40% of the exposed participants had a diagnosis of major depression with suicide risk. Regarding nonexposed participants, no psychiatric disorders were detected in 64% of the population. Environmental exposure to OPs may be associated with depression and suicide attempts in populations living in a rural agricultural area, since high suicide rates were reported in these areas [8,36,37]. Suicide is a complex phenomenon, and may be the result of an interaction of multiple biological, psychological, and socioeconomic factors, as well as environmental pollutants [38]. Further studies are necessary to elucidate the factors involved in the high suicidal risk found in the exposed participants, to stablish suicide-prevention policies. The high suicide risk of exposed females requires particular attention.

The average hemoglobin content for females of both exposed and nonexposed participants was typically considered as normal for the Mexican population, according to Diaz et al. [22]. Men from the exposed participants had a lower hemoglobin content than the nonexposed men. In this study, hemoglobin was lower in the exposed participants than in the nonexposed group. These results agree with those described by Rojas-García et al. [39], who found a significant decrease in the mean hemoglobin, hematocrit, and red blood count in 83 male pesticide retailers from Nayarit, a relevant agricultural region in Mexico. However, we currently do not have an explanation of why we only found a reduction in the hemoglobin of exposed males, but not in exposed females. A decrease in AChE activity was observed in several studies after OP exposure [8,40,41,42]. In this study, the mean value of AChE enzymatic activity, the specific activity for the exposed participants was lower than the enzymatic activity of the nonexposed participants. The decrease in AChE activity was thought to be related to the duration and frequency of pesticide exposure [42]. In this study, a relationship was found between AChE activity and the span of the exposure, using the number of years working in the field (~3 years average). Ramirez-Santana et al. [42] evaluated AChE activity in 87 occupationally exposed workers on grapes and citrus-fruit plantations. The farm workers were evaluated throughout the course of one year to cover the prespraying season and the spraying season during the harvest season in Chile’s Coquimbo region (Pacific Coast). It was reported that 41% of the occupationally exposed group demonstrated AChE inhibition. 

Organophosphate-inhibited AChE leads to the overstimulation of muscarinic and nicotinic cholinergic receptors [2,3]. Long-term symptoms of OP-induced chronic neurotoxicity caused by exposure to OP pesticides, and symptoms of acute cholinergic toxicity, are of clinical relevance as they can appear with a number of signs and symptoms that present based on types of cholinergic receptors, such as tremors, muscle weakness, mydriasis tachycardia, hypertension, sweating, weakness, fasciculations in nicotinic peripheral effects; salivation, urination, lacrimation, gastrointestinal pain, bronchospasm, anxiety, emotional lability, insomnia, numbness, anorexia, fatigue, lethargy, restlessness, depression, headaches, and muscarinic peripheral effects [43,44,45,46]. A possible mechanism that may lead to delayed neurotoxicity includes damage to the afferent fibers of peripheral and central nerves associated with the irreversible inhibition of AChE, resulting in delayed brain damage. Delayed secondary neuronal destruction, which arises primarily in the cholinergic areas of the brain that contain dense accumulations of cholinergic neurons and the majority of cholinergic projections, could be responsible for persistent profound neuropsychiatric and neurological impairments, such as memory, cognitive, mental, emotional, motor, and sensory deficits in victims of OP poisoning [47,48].

Most previous studies of pesticides and neurologic symptoms have focused on OP pesticides. The percentage of AChE activity inhibition required to produce a clinical manifestation is as follows: normal functioning is expected if inhibition is less than or equal to 25%, mild inhibition should occur if the activity ranges from 26–70%, moderate inhibition is expected to occur if it is in the range of 71–90%, and severe inhibition tends to be observed if the activity is < 10% (47). Our results show that, with 25% inhibition (Table 4 and Table 5), greater clinical manifestations were observed, such as major depression with suicide risk and farmer’s syndrome (muscarinic peripheral effects), in this group of patients, along with a higher incidence of back pain and dizziness. Neurological symptoms are a rural-area health problem that commonly affects farmer workers [30]. The current study demonstrated that the presence of neurological symptoms significantly elevates the risk of suicidal ideation. This is a significant finding considering our results that have shown that the group with the highest suicide risk had the highest percentage of AChE inhibition, supporting the association between exposure to pesticides and the risk of suicide.

Our results also agree with the fact that neuropsychiatric manifestations are not observed in patients with the lowest degree of AChE inhibition (22%), demonstrating that this study supports the relationship between psychiatric disorders and exposure to OPs.

This study has some limitations that could be remedied with future research. For example, we did not include the determination of pesticides through the levels of metabolites in urine. The state of exposure to OPs was verified using sociodemographic data, obtained through a questionnaire. This does not allow the accurate classification of our sample according to the degree of OP exposure, which may have resulted in exposure misclassification. Another limitation is that age and education were not homogeneous between the groups. Other confounding factors, such as tobacco, alcohol, and drug use, did not show differences between groups. Therefore, analysis in this report considers age and education as confounding factors, as described in the literature. Despite these limitations, our data provide crucial information, such as the differences in AChE levels, as well as the enzymatic activity corresponding to different states of mental health in the studied populations.

## 5. Conclusions

The present study clearly showed that chronic exposure to pesticides among agricultural workers causes a considerable decrease in red blood AChE activity, which generates neuropsychiatric disorders ranging from generalized anxiety to high suicide risk. Our results indicate that pesticide exposure calls for the development of effective public-health strategies. We recommend intensive health education and training for farm workers on the use of agrochemicals to prevent serious health complications that can occur because of their inappropriate use.

## Figures and Tables

**Figure 1 ijerph-16-00689-f001:**
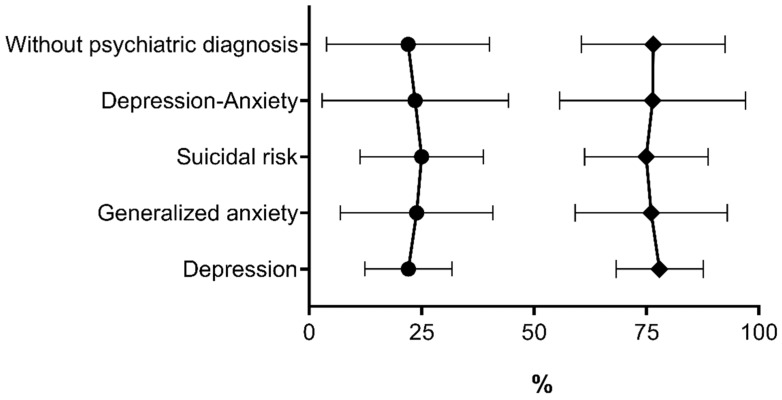
Comparison levels of % inhibition (•) and AChE activity (♦)% of the exposed participants. Data are represented as the mean ± SD.

**Table 1 ijerph-16-00689-t001:** Sociodemographic characteristics of the general study population. *p*-values were obtained with the Chi-Square test. *p* < 0.05 was considered statistically significant.

Variable	Nonexposed (*n* = 100)		Exposed (*n* = 140)		*p*
	*N*	%	*N*	%	
Age (Years)					0.000
18–34	89	89.0	84	60	
35–50	9	9.0	52	37.1	
>51	2	2.0	4	2.9	
Gender					0.751
Female	60	60.0	88	62.9	
Male	40	40.0	52	37.1	
Education					0.000
Illiterate	0	0	12	8.6	
Elementary school	1	1.0	78	55.7	
Middle school	2	2.0	38	27.1	
High school	0	0	10	7.1	
College	97	97	2	1.4	
Marital status					0.000
Single	60	60.0	20	14.3	
Free union	35	35.0	68	48.6	
Married	4	4.0	48	34.3	
Divorced	1	1.0	0	0	
Widowed	0	0	4	2.9	
Occupation					0.000
Farmworker	20	20.0	140	100.0	
Student	80	80.0	0	0	
Place of residence in the last six months					0.000
Provincial center	100	100.0	0	0	
Surrounding village	0	0	140	100.0	
Smokers	23	23.0	22	15.7	0.040
Alcohol drinkers	54	54.0	24	17.4	0.000
Abuse substances					0.190
Cocaine	3	3.0	4	2.8	
Heroin	0	0	3	2.1	
Marihuana	5	5.0	12	8.6	
Crystal	1	1.0	11	7.9	
No abuse substances	91	91.0	110	78.6	
Diagnosed psychiatric disorder					0.000
Depression	3	3.0	20	14.3	
Major depression with suicidal risk	8	8.0	44	31.4	
Generalized anxiety	18	18.0	20	14.3	
Depression–generalized anxiety	7	7.0	20	14.3	
No psychiatric-disorder diagnosis	64	64.0	36	25.7	

**Table 2 ijerph-16-00689-t002:** Acethylcholinestease activity data of the general study population. The *p*-values for each parameter were obtained by factorial ANOVA, and *p* < 0.05 was considered statistically significant.

Parameter	Nonexposed (Mean ± SD)	Exposed (Mean ± SD)	*F*	*p*
Hb (g/dL)			21.8	0.000 *
Female	12.72 ±1.07	12.66 ± 2.25		
Male	15.15 ± 0.90	13.74 ± 2.54		
AChE (mU/μLHb)			33.2	0.000 *
Female	582.43 ± 53.63	435.69 ± 103.28		
Male	485.39 ± 40.78	398.99 ± 84.89		
AChE activity (mE/min)	227.95 ± 10.19	178.88 ± 37.48	126.9	0.000
AChE activity (%)	97.53 ± 4.33	76.39 ± 15.93	126.9	0.000
AChE inhibition (%)	2.4 ± 4.33	23.32 ± 15.92	126.9	0.000

* The interaction between sex and group (exposed vs. unexposed) was significant at the 0.05 level.

**Table 3 ijerph-16-00689-t003:** Organophosphate pesticides commonly used by the study population. World Health Organization classification: Class IA, extremely hazardous; Class IB, highly hazardous; Class II, toxic; Class III, moderately toxic.

Pesticide Name	WHO * Classification
Azinphos-methyl	IB
Chlorpyrifos	III
Dicofol	II
Malathion	III
Methamidophos	IB
Methyl parathion	IA
Phosmet	II

* World Health Organization.

**Table 4 ijerph-16-00689-t004:** Risk of toxicity and neurological symptoms of the exposed participants diagnosed with psychiatric disorders. *p*-values were obtained by the Chi-Square test. *p* < 0.05 was considered statistically significant. The percentage was calculated concerning the exposed participants. Percentages were determined concerning the total of the exposed participants.

Variable (*N*) %	Depression Disorder	Suicidal Attempt	Generalized Anxiety	Depression and Generalized Anxiety	No Psychiatric Diagnosis Disorder	*p*
Weekly pesticide exposure time in hours (*N*)						
30–60	(16) 11.4	(40) 28.6	(14) 10.0	(20) 14.3	(32) 22.9	0.045
61–90	(4) 2.9	(4) 2.9	(6) 4.3		(4) 2.9	
Annual pesticide-exposure time in years (*N*)						
0.5–1	(12) 8.6	(22) 15.7	(12) 8.6	(10) 7.1	(22) 15.7	0.113
2–9	(8) 5.7	(20) 14.3	(6) 4.3	(10) 7.1	(8) 5.7	
10–20	(0)0	(2) 1.4	(2) 1.4	(0) 0	(6) 4.3	
Neurological symptoms (*N*)						
Shoulder pain	(6) 4.3	(28) 20	(2) 1.4	(14) 10	(18) 12.9	0.001
Back pain	(10) 7.1	(34) 24.3	(10) 7.1	(18) 12.9	(18) 12.9	0.003
Numbness	(6) 4.3	(24) 17.1	(8) 5.7	(14) 10	(20) 14.3	0.095
Nocturia	(4) 2.9	(18) 12.9	(6) 4.3	(6) 4.3	(14) 10	0.504
Dyspnea	(4) 2.9	(16) 11.4	(4) 2.9	(10) 7.1	(10) 7.1	0.175
Insomnia	(2) 1.4	(12) 8.6	(4) 2.9	(8) 5.7	(10) 7.1	0.267
Dizzines	(10) 7.1	(30) 21.4	(2) 1.4	(14) 10	(16) 11.4	0.001
Abdominal discomfort	(8) 5.7	(23) 16.4	(0) 0	(12) 8.6	(14) 10	0.001

**Table 5 ijerph-16-00689-t005:** Acethylcholinesterase activities of the study population diagnosed with different psychiatric disorders. The *p*-values for each parameter were obtained by factorial ANOVA, and *p* < 0.05 was considered statistically significant.

Psychiatric Disorders	Acethylcholinesterase Activity Values			
	Specific Activity, mU/μLHb (Mean ± SD)	Activity, mE/min (Mean ± SD)	% Activity (Mean ± SD)	% Inhibition (Mean ± SD)
Depression disorder				
Exposed	436.49 ± 85.66	181.85 ± 22.85	77.94 ± 9.70	22.06 ± 9.71
Nonexposed	481.31 ± 117.60	208.93 ± 33.71	89.45 ± 14.32	10.55 ± 14.32
*p*	0.377	0.057	0.057	0.057
Major depression with suicidal risk				
Exposed	414.34 ± 92.75	174.89 ± 32.33	74.99 ± 13.74	25.01 ± 13.73
Nonexposed	555.86 ± 55.25	228.43 ± 7.27	97.73 ± 3.08	2.27 ± 3.09
*p*	0.000	0.000	0.000	0.000
Generalized anxiety				
Exposed	388.62 ± 90.04	177.48 ± 39.87	76.08 ± 16.94	23.91 ± 16.94
Nonexposed	553.44 ± 72.42	228.11 ± 7.34	97.60 ± 3.12	2.340 ± 3.12
*p*	0.000	0.000	0.000	0.000
Depression and generalized anxiety				
Exposed	421.66 ± 106.27	178.23 ± 48.70	76.40 ± 20.69	23.59 ± 20.69
Nonexposed	566.99 ± 78.36	225.71 ± 8.20	96.58 ± 3.49	3.42 ± 3.49
*p*	0.003	0.000	0.000	0.000
Without a psychiatric diagnosis				
Exposed	440.25 ± 108.81	181.96 ± 42.73	76.54 ± 15.93	22.01 ± 18.15
Nonexposed	539.75 ± 63.44	229.28 ± 7.65	98.09 ± 3.25	1.91 ± 3.25
*p*	0.000	0.000	0.000	0.000

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
