# Peer review of "Neuropsychiatric Disorders in Farmers Associated with Organophosphorus Pesticide Exposure in a Rural Village of Northwest México"

_ijerph, 2019, doi:10.3390/ijerph16050689_

Round 1

Reviewer 1 Report

This is a thorough and timely study on the manifestation of neuropsychiatric disorders in farmers exposed to organophosphorus pesticides in Baja California, Mexico. The manuscript is well-written and of interest to the general readership. I recommend publishing in the present form.

Author Response

We thank the reviewer for their kind comments.

Reviewer 2 Report

The authors studied the “Neuropsychiatric Disorders in Farmers Associated with Organophosphorus Pesticide Exposure in a Rural Village of Northwest México”. While I have no objection against publishing the data, I have some issues that need addressing. I am concerned about the extensive repetition of elements of the introduction section (should be summarized). The experimental set up of this study appears to be well-designed and the data collected carefully. However, information about the P values obtained in the statistical analysis must be revised and provide other statistical values (F for ANOVA). I think that this manuscript requires substantial rewriting to make its results clearer and more readily interpretable to the reader. My specific comments are listed in the "Main text". Based on the comments above reported, my opinion is that this manuscript may be suitable for printing on this journal after corrections.

Author Response

We thank the comments of the reviewer. Their suggestions have been now considered in the manuscript. The introduction section has been summarized. The P values have been revised. The F values for ANOVA have been included. The specific comments in Main text have been considered. The manuscript has been rewritten and proofread by a native English speaker.

Reviewer 3 Report

I appreciate the contribution that this study makes to the literature to help associate OP pesticide exposure and neuropsychiatric disorders. I have a few major and minor concerns that I think should be considered prior to publication.

Major

More needs to be shared about the MINI- validity and reliability information.

I am not sure if I would refer to any of the mental status categories as a diagnosis. While psychiatry is not my area of expertise is the MINI ever used for true diagnoses? Can medical students make diagnoses? If not, I would steer clear of that language and refer to this as results of the MINI/ scoring of the MINI or something similar.

I think it is more than a minor limitation that the control group was so different from the exposed group. Significant differences included not only age and education but also marital status. More should be shared about how the control group was chosen. Were these college students chosen by convenience? If so that should be explicitly stated and more should be said about the third confounder (marital status) in the discussion as well as a discussion of how these factors might affect the outcomes. Also, I was unclear- was the data collected from the control group over three time periods? If not, why not? Was there anything that occurred over those 3 time periods that might have made a difference in results? The paragraph that spans line 293-303 is way overstated given those confounders.

Differences in gender were pointed out in results but not really discussed.

The sentence in the conclusion- line 366- that exposure to pesticides occurs from inadequate education, training, and safety systems- does not draw at all from your own findings.

Minor

The English language requires editing. In particular, attention to singular v. plural and tense should be given. There are also terms that read awkwardly (limits to the south with- line 55, stands out for- line 59,

I suggest that the word participants be used throughout the paper. The following terms are used interchangeably and cause confusion: participants, volunteers, exposed and non-exposed populations, exposed and non-exposed workers.

Line 16-19 is a long sentence which is awkwardly worded.

I recommend stating the name of the validated(?) tool – the MINI- even in the abstract.

Please move some of the information from the discussion about enzymatic activity into the introduction. I think it would have helped me to read that prior to reading the results.

On the other hand is used a couple of times (Line 53, Line 293) when the information presented is not in contrast to what preceded it. This should be remedied.

If the mother language is Spanish- line 94- say so, or was more than one language offered.

I didn’t know until sample preparation that signed consent forms were kept by the study team- maybe move up to ethical statement and describe storage.

Line 98- might it be more accurate to say- would not be penalized, and would receive no payment for their participation?

Line 107- is it more accurate to say the med students administered and scored and did not diagnose?

I might consider reporting only to one percentage point in the results for demographic data. I might also call the MINI by it’s name instead of the face-to-face interview.

I don’t understand the short paragraph that is lines 174-178. How were females considered normal and still significantly different from each other in terms of exposed and non-exposed participant groups?

Line 218- I suggest changing determined according to assessed relative to.

Thank you for the opportunity to review this manuscript. I wish the authors success in their efforts to publish this manuscript in IJERPH.

Author Response

More needs to be shared about the MINI- validity and reliability information.

R. Done.

I am not sure if I would refer to any of the mental status categories as a diagnosis. While psychiatry is not my area of expertise is the MINI ever used for true diagnoses? Can medical students make diagnoses? If not, I would steer clear of that language and refer to this as results of the MINI/ scoring of the MINI or something similar.

R. The MINI is, a short diagnostic interview schedule that can be easily incorporated into routine clinical interviews; it is a detection instrument and diagnostic orientation in the discipline of psychiatry and psychology. It is feasible, acceptable and reliable test-retest  [18, 19]. The med students administrated the MINI. The mental state of the participants was determined by a Psychologist (member of the study team), from the result of the MINI. This is now clarified in the manuscript.

I think it is more than a minor limitation that the control group was so different from the exposed group. Significant differences included not only age and education but also marital status. More should be shared about how the control group was chosen. Were these college students chosen by convenience? If so that should be explicitly stated and more should be said about the third confounder (marital status) in the discussion as well as a discussion of how these factors might affect the outcomes. Also, I was unclear- was the data collected from the control group over three time periods? If not, why not? Was there anything that occurred over those 3 time periods that might have made a difference in results? The paragraph that spans line 293-303 is way overstated given those confounders.

R. The non-exposed participants were chosen by convenience, in the same three periods than the exposed participants. This is now stated in the manuscript. To our knowledge, this is the first study to assess the mental state of farm workers in northern Mexico. I high incidence of depression with suicidal risk was found. More studies are needed to determine all the causes of those health issues. The study indicates the there is a need to stablish health programs to prevent and treat mental diseases. However, the control group was very important to show that AChE activities are reduced in the exposed participants. We did not observe differences in the sampling periods, but the three sampling periods were convenient for the size of the samples. The paragraph lines 293-303 have been rewritten to avoid overstatement.

Differences in gender were pointed out in results but not really discussed.

R. Differences in gender are now discussed.

The sentence in the conclusion- line 366- that exposure to pesticides occurs from inadequate education, training, and safety systems- does not draw at all from your own findings.

R. The statement was eliminated since it is not drawn from our findings.

Minor

The English language requires editing. In particular, attention to singular v. plural and tense should be given. There are also terms that read awkwardly (limits to the south with- line 55, stands out for- line 59,

R. The manuscript was proofread by a native English speaker. The paragraph has been eliminated to summarize the introduction.

I suggest that the word participants be used throughout the paper. The following terms are used interchangeably and cause confusion: participants, volunteers, exposed and non-exposed populations, exposed and non-exposed workers.

R. The term participants is now used throughout the manuscript.

Line 16-19 is a long sentence which is awkwardly worded.

R. Reworded

I recommend stating the name of the validated(?) tool – the MINI- even in the abstract.

R. Done.

Please move some of the information from the discussion about enzymatic activity into the introduction. I think it would have helped me to read that prior to reading the results.

R. Done.

On the other hand is used a couple of times (Line 53, Line 293) when the information presented is not in contrast to what preceded it. This should be remedied.

R. Remediated.

If the mother language is Spanish- line 94- say so, or was more than one language offered.

R. Stated.

I didn’t know until sample preparation that signed consent forms were kept by the study team- maybe move up to ethical statement and describe storage.

R. Done.

Line 98- might it be more accurate to say- would not be penalized, and would receive no payment for their participation?

R. Changed.

Line 107- is it more accurate to say the med students administered and scored and did not diagnose?

R. Changed.

I might consider reporting only to one percentage point in the results for demographic data. I might also call the MINI by it’s name instead of the face-to-face interview.

R. Done.

I don’t understand the short paragraph that is lines 174-178. How were females considered normal and still significantly different from each other in terms of exposed and non-exposed participant groups?

R. There was no significative difference in the hemoglobin content between exposed and non-exposed females. The p values have been corrected. The difference is in the male groups. It is now clarified in the manuscript.

Line 218- I suggest changing determined according to assessed relative to.

R. To avoid confusion the sentence was eliminated, which make the manuscript more readable.

Thank you for the opportunity to review this manuscript. I wish the authors success in their efforts to publish this manuscript in IJERPH.

R. We thank you for your insightful comments and suggestions.